# Influences on university students' intention to receive recommended vaccines: a cross-sectional survey

Kate Landowska,[1] Jo Waller,[1] Helen Bedford,[2] Lauren Rockliffe,[1] Alice S Forster[1]

► Prepublication history and additional material are available. To view, please visit the journal online (http://dx.doi.org/10.1136/bmjopen-2017-016544).

## ABSTRACT

**Objectives** To explore predictors of university students' intention to receive a recommended vaccine and the main sources of vaccine-related information accessed by university students.

**Setting** Participants were recruited from University College London (UK) in summer 2015.

**Participants** 177 university students participated. The majority of participants were female (58%), White (68%) and had no religion (58%). Participants were aged 18 to 42 (mean age=23.6).

**Primary and secondary outcome measures** Primary outcome measures included vaccine attitude, perceived subjective norm, perceived behavioural control, perceived self-efficacy, past receipt of recommended childhood vaccines, perceived adverse reaction to past vaccination and needle fear. As a secondary outcome sources of vaccine-related information were assessed.

**Results** Students classified as high intenders were more likely to have received all recommended childhood vaccines (OR 3.57; 95% CI 1.21 to 10.59; p=0.022), be less afraid of needles (OR 2.44; 95% CI 1.12 to 5.36; p=0.026) and to have lived in the UK until at least the age of 4 compared with those not living in the UK until at least the age of 4 (OR 0.39; 95% CI 0.18 to 0.83; p=0.015) and those who lived both in the UK and elsewhere (OR 0.42; 95% CI 0.04 to 4.06; p=0.424). The multivariable model explained 25.5% of variance in intention to receive a recommended vaccine. The internet was the most commonly reported source of vaccination information.

**Conclusions** Findings provide an indication of the factors that may need to be addressed by interventions aiming to increase uptake of recommended vaccines in a university population. Future research is recommended using a prospective cohort design.

## Strengths and limitations of this study

► This study was sufficiently powered to identify predictors of intention to vaccinate in a university student population.

► It was not possible to determine the response rate to the survey; participants may have been those with an interest in the subject.

► Findings are correlational so future research should consider using a prospective design.

numbers of people remain unprotected, disease outbreaks can occur (eg, Iacobucci[2]). Understanding why some people do not get vaccinated is complex. Some people appear to reject vaccines without question, while others experience uncertainty, which might result in a delay in obtaining the vaccine or rejection of the vaccine.[3–6] Practical barriers can also prevent immunisation.

Previous research has suggested that vaccine uptake is significantly predicted by an individual's intention to vaccinate.[7] The Theory of Planned Behaviour[8] and the Health Belief Model[9] have both been shown to predict intention to vaccinate.[10] Other predictors include perceived self-efficacy (PSE; a person's belief about their capability of accomplishing a task)[11], previous vaccination experience[12] and needle fear.[13]

There has been little research investigating predictors of vaccine intention among students attending UK universities. It is important to understand how to promote vaccine uptake in this population as in the UK vaccines are recommended specifically for this group if they are unprotected when entering university. The main vaccination recommended for university entrants since 2015 is the Meningococcal ACWY (MenACWY) vaccination, which protects against life-threatening meningitis and septicaemia and was introduced in response to the rapid increase in MenW cases. Rates of nasopharyngeal carriage of meningococcal

## INTRODUCTION

Vaccine-preventable diseases (VPDs) are associated with significant mortality and morbidity. Certain groups of individuals are more likely to be exposed to particular VPDs. For example, university students are at higher risk of meningitis and septicaemia due to living in close contact in shared accommodation.[1] It is important to maintain high vaccine coverage to control incidence of VPDs and protect high-risk groups. If sufficient

[1]Research Department of Behavioural Science and Health, University College London, London, UK
[2]Great Ormond Street Institute of Child Health, University College London, London, UK

**Correspondence to**
Dr Alice S Forster; alice.forster@ucl.ac.uk

organisms are particularly high in university students, and the close contact they have with one another due to their living arrangements and lifestyle puts them at increased risk of disease.[14] It is also important that students have had two doses of the measles, mumps and rubella (MMR) vaccine. Recent outbreaks of measles in England have mainly affected unimmunised teenagers and young adults who may have missed out on the vaccine at the height of public concerns over its safety in the early 2000s.[15 16]

In this study, we carried out a survey to identify (1) predictors of university students' intention to receive a recommended vaccine and (2) their main sources of vaccine-related information.

## METHODS

This study was conducted between June and July 2015. One hundred and seventy-seven students were recruited via an advertisement email circulated by administrators in 39 departments at University College London (UCL), based on a sample size calculation ($f^2$=0.1[17]; alpha=0.05; power=0.8; predictors=10). This calculation suggested that a minimum sample size of 91 participants would be required for the study to be sufficiently powered. Potential participants were given the incentive of being entered into a prize draw to win an iPad. The study was approved by the UCL research ethics committee (6613/001).

Participants completed a 48-item online survey (see online supplementary file). Items were developed from previous surveys.[18–20] These surveys were selected as they investigated predictors (ie, Theory of Planned Behaviour constructs) of vaccine acceptance and positive behaviour change. Intention to receive a recommended vaccine was measured using two items, responses to which were summed to give a possible score of 2–10; 'I would try to get a recommended vaccine' and 'I would intend to get a recommended vaccine' (five-point scale: strongly disagree[1] to strongly agree[5]). Predictor variables measured included vaccine attitude, perceived subjective norm (PSN; whether the respondent thinks that significant others would approve of them getting a vaccine and their motivation to comply with these wishes), perceived behavioural control (PBC; perceived ease in getting a vaccine), PSE, past receipt of recommended childhood vaccines, perceived adverse reaction to past vaccination and needle fear. Participants also provided demographic information. Sources of vaccine-related information were assessed using a single item, modified from the Health Information National Trends Survey, 2013.[21] Participants were not able to proceed with the survey if they failed to provide a response to an item. This ensured no data were missing.

Informed consent was gained from all participants at the start of the survey by means of an online consent form. Participants could only proceed to complete the survey once the consent form had been completed.

Data were analysed using SPSS V.19. The score for intention to receive a recommended vaccine was skewed, so a binary outcome was created using a median split, with those scoring 2–9 (out of 10) classified as 'low intenders' and those scoring 10 as 'high intenders'. Predictor variables were recoded into binary variables using a median split including attitude (positive/negative), PSN, PBC and PSE (all high/low) and needle fear (scared/not scared). Response options for past vaccine receipt was grouped into 'not all childhood vaccines received' or 'all childhood vaccines received'.

A logistic regression analysis was performed to examine the association between predictor variables and vaccine intention. All factors showing significant associations with intention on univariable analyses (p<0.05) were entered into a multivariable model. For the secondary question, we tabulated the number of students who reported using each source of vaccine-related information. As a sensitivity analysis we reran the original analysis splitting intention at the median, minus the median absolute deviation (2–8 vs 9–10).

## RESULTS

### Participants

A total of 177 students completed the vaccine intention questions (57.6% female; 37.9% male; 4.5% prefer not to say/unknown; aged 18–42 (mean age=23.61, SD=4.43)). The majority of participants were White (68%), followed by Asian (16%), Black (3%), mixed ethnicity (4%) and other (3%). Most participants self-defined as having no religion (58%), followed by Christian (24%) and other religions (12%). Approximately 44% had lived outside the UK until at least the age of 4, the period in which childhood vaccinations are offered. This is somewhat higher than the proportion of international students attending UCL, which is 35%. It was not possible to estimate a response rate due to the method of contacting potential participants.

### Predictors of vaccine intention

Factors significantly associated with intention were entered into a multivariable model, which explained 25.5% of variance in intention to receive a recommended vaccine (table 1). Past vaccine receipt, needle fear and country of residence until at least the age of 4 remained significant predictors of vaccine intention in the model (p=0.022, p=0.026 and p=0.015, respectively). Participants who had received all recommended childhood vaccines were over three times more likely to be high vaccine intenders than participants who had not (OR 3.57, 95% CI 1.21 to 10.59). Students who were less afraid of needles were over two times more likely to be high vaccine intenders than students who were more afraid (OR 2.44, 95% CI 1.12 to 5.36). Moreover, participants who did not live in the UK until the age of four were significantly less likely to be high vaccine intenders than participants who had (OR 0.39, 95% CI 0.18 to 0.83).

When we reran the analyses dichotomising intention at 8 rather than 9, the results were very similar with no change in significance (analyses not shown).

**Table 1** Behavioural and demographic predictors of having a high intention to receive a recommended vaccine (n=177*)

| Variable | n (%) with high intention to receive a recommended vaccine† | Univariable analyses OR (95% CI) p | Multivariable analysis (n=177*) OR (95% CI) p |
|---|---|---|---|
| Attitude | | | |
| Negative attitude | 25 (27.5) | 1.00 | 1.0 |
| Positive attitude | 54 (62.8) | 4.46 (2.36 to 8.41)≤0.001 | 1.74 (0.75 to 4.05) 0.196 |
| PSN | | | |
| Low PSN | 28 (30.4) | 1.00 | 1.0 |
| High PSN | 51 (60.0) | 3.43 (1.84 to 6.38)≤0.001 | 2.22 (0.92 to 5.35) 0.077 |
| PBC | | | |
| Low PBC | 40 (40.4) | 1.00 | |
| High PBC | 39 (50.0) | 1.48 (0.81 to 2.68) 0.203 | |
| PSE | | | |
| Low PSE | 29 (32.6) | 1.00 | 1.0 |
| High PSE | 50 (56.8) | 2.72 (1.48 to 5.02) 0.001 | 1.07 (0.46 to 2.51) 0.876 |
| Past receipt of recommended childhood vaccines | | | |
| Not all recommended childhood vaccines | 6 (15.8) | 1.00 | 1.0 |
| All recommended childhood vaccines | 72 (53.3) | 6.1 (2.39 to 15.53)≤0.001 | 3.57 (1.21 to 10.59) 0.022 |
| Perceived adverse reaction to past vaccination | | | |
| Bad reaction | 4 (22.2) | 1.00 | 1.0 |
| No bad reaction | 72 (47.7) | 3.19 (1.00 to 10.14) 0.049 | 3.53 (0.92 to 13.53) 0.066 |
| Needle fear | | | |
| Scared of needles | 40 (36.0) | 1.00 | 1.0 |
| Not scared of needles | 39 (59.1) | 2.56 (1.37 to 4.79) 0.003 | 2.44 (1.12 to 5.36) 0.026 |
| Age | | | |
| 18–25 | 59 (46.8) | 1.00 | |
| 26–33 | 15 (39.5) | 0.74 (0.35 to 1.55) 0.426 | |
| 34–42 | 2 (28.6) | 0.45 (0.09 to 2.43) 0.356 | |
| Gender | | | |
| Male | 27 (40.3) | 1.00 | |
| Female | 49 (48.0) | 1.37 (0.73 to 2.56) 0.323 | |
| Ethnicity | | | |
| White British/Other | 59 (48.8) | 1.00 | |
| Asian British/Other | 10 (34.5) | 0.55 (0.24 to 1.29) 0.169 | |
| Black British/Other | 0 (0) | – | |
| Mixed ethnicity British/Other | 4 (50.0) | 1.05 (0.25 to 4.40) 0.946 | |

Continued

**Table 1** Continued

| Variable | n (%) with high intention to receive a recommended vaccine† | Univariable analyses OR (95% CI) p | Multivariable analysis (n=177*) OR (95% CI) p |
|---|---|---|---|
| Other ethnic group | 1 (16.7) | 0.21 (0.02 to 1.85) 0.160 | |
| Country of residence until the age of 4 | | | |
| UK | 51 (58.0) | 1.00 | 1.0 |
| Not UK | 23 (29.9) | 0.31 (0.16 to 0.59) ≤0.001 | 0.39 (0.18 to 0.83) 0.015 |
| Both | 2 (33.3) | 0.36 (0.06 to 2.09) 0.256 | 0.38 (0.04 to 4.06) 0.424 |
| Religion | | | |
| No religion | 47 (46.1) | 1.00 | |
| Christian | 18 (41.9) | 0.84 (0.41 to 1.73) 0.641 | |
| Other religion | 9 (40.9) | 0.81 (0.32 to 2.06) 0.659 | |
| Deprivation (possible range: 2–20) | Mean=9.78, SD=2.97 | 1.11 (1.00 to 1.23) 0.060 | |

*Note that n differs slightly between items due to missing data.
†Unless specified.
PBC, perceived behavioural control; PSE, perceived Self-efficacy; PSN, perceived subjective norm.

**Table 2** Sources of vaccine-related information (n=177*)

| Source | n (%) that has used source for vaccine-related information |
|---|---|
| Internet | 118 (66.67) |
| Doctor or healthcare provider | 100 (56.50) |
| Family | 44 (24.86) |
| Brochures, pamphlets | 42 (23.73) |
| Friend/coworker | 26 (14.69) |
| Books | 6 (3.39) |
| Newspapers | 6 (3.39) |
| Library | 4 (2.26) |
| Magazines | 4 (2.26) |
| Telephone information number | 4 (2.26) |
| Complementary, alternative or unconventional practitioner | 2 (1.13) |
| Other (examples reported: university student psychological services; organic and ethical health forums; pharmacy; someone with a similar health problem) | 3 (1.69) |

*Column n>177 as multiple responses were permitted.

### Information sources

The five most commonly reported vaccine information sources were the internet (66.7%); doctor or healthcare provider (56.5%); family (24.9%); brochures or pamphlets (23.7%) and friend or coworker (14.7%). All other sources were reported by <4% of students (table 2). The most commonly specified websites were the National Health Service (NHS), search engines (eg, Google) and Health/Medical information websites (eg, webMD; patient.co.uk).

### DISCUSSION

This study explored influences on university students' intentions to receive a recommended vaccine. We found that past vaccine receipt, needle fear and country of residence until at least the age of four were significant predictors of vaccine intention.

Past vaccine receipt had the strongest influence on intention. Students who had received all recommended childhood vaccines were over three times more likely to be high vaccine intenders than students who had not. This is consistent with a previous systematic review which found that receiving all childhood vaccinations was a positive predictor for human papillomavirus vaccine initiation in teenage girls.[12]

Furthermore, students who were less afraid of needles were more than twice as likely to be high vaccine intenders than students who were more afraid. This supports a previous study which reported needle fear to be a primary reason for vaccine avoidance among adolescent boys.[13] Interventions addressing vaccine uptake in a

student population could therefore benefit from focusing on psychological targets such as reducing needle fear by using interventions to reduce pain on immunisation. Moreover, interventions could target students who did not complete their childhood immunisation schedules.

Participants who did not live in the UK before the age of 4 were significantly less likely to be high vaccine intenders compared with participants who had. To our knowledge, there is no previous research investigating the influence of country of childhood residence on intention to vaccinate as an adult. This could be an interesting direction for future research which could assess differences in immunisation schedules between countries and the effects of disparities on uptake.

The five most commonly reported sources of vaccine-related information were the internet, doctor/healthcare provider, family, brochures/pamphlets and friend/coworker. The popularity of these sources is consistent with a critical appraisal reviewing 38 studies.[22] The findings of this study indicate that the most common vaccine-related source for students is the internet, which corroborates previous research which found the internet to be the most common health and medicine source used in the UK.[23] However, previous studies suggest that the internet is perceived as less trustworthy than sources such as a general practitioner and a medically qualified friend or relative.[23] It is therefore unlikely to supplant the role of these sources, but may be used to supplement them.[24] Official websites (ie, NHS website, Web MD) were reported more frequently than blogs and Wikipedia. This substantiates previous findings suggesting that official websites are trusted more than less well-moderated websites.[22] No students reported social media as a source of online vaccine-related health information. Campaigns could therefore benefit from disseminating information on official web pages. Additionally, university students in the UK often live away from home so re-register with a local health provider. Healthcare providers should use this opportunity to provide information in consultations and use pamphlets as a tool to offer further information.

There are several limitations to this study. Due to the cross-sectional design, findings are correlational. It is therefore difficult to infer the direction of causality between outcome and predictor variables. A future prospective cohort design investigating factors predicting vaccine uptake could therefore be insightful. Intention to vaccinate in the present study may not be directly predictive of behaviour. Previous research has demonstrated an intention–behaviour gap.[7] Moreover, we cannot be sure of the response rate as it is not clear how many university departments circulated the recruitment email. Consequently, we are unable to assess non-response bias and there may have been a response bias towards students more interested in this subject. Some survey items relied on participants' knowledge of vaccinations received during childhood and perception of past adverse reactions, which may also have occurred during childhood. There is therefore the potential for inaccuracies in participant reporting, though it could be argued that perceptions of vaccine-related events in childhood would be stronger predictors of subsequent vaccine attitudes than objectively measured events. In addition, the high proportion of participants who lived outside the UK until the age of 4 suggests an over-representation of non-UK-born participants. It is therefore uncertain whether the findings would be directly generalisable to the wider student population in the UK. Finally, due to time restrictions, we were unable to pilot the survey prior to data collection and there is the potential that some items may lack clarity. A strength of the study is that the sample was large and so we can draw useful conclusions about this sample.

These findings are timely in view of the recent introduction of MenACWY vaccine for 14–18-year olds and for young people up to the age of 25 years attending university for the first time. It is of concern that uptake of the MenACWY vaccine in the first two cohorts of 18-year olds offered the vaccine has not exceeded 39%, leaving many young people at risk of these potentially severe infections.[25] The findings provide an indication of where to focus future interventions to raise university students' awareness of the importance of this and other vaccination programmes.

**Contributors** KL, AF, JW and HB developed the study idea and design. KL collected study data and conducted the analysis. AF, JW, HB and LR assisted with data analysis. KL, AF, JW, HB and LR contributed to the write-up and interpretation of study data.

**Funding** AF and LR are funded by Cancer Research UK—BUPA cancer prevention fellowship awarded to AF (C49896/A17429). JW is funded by Cancer Research UK Career Development Fellowship (C7492/A17219).

**Competing interests** None declared.

**Patient consent** Obtained.

**Ethics approval** UCL Research Ethics Committee.

**Provenance and peer review** Not commissioned; externally peer reviewed.

**Data sharing statement** Data are available on request from the corresponding author.

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
