## [Reviewer comments · BMJ Open]

ARTICLE DETAILS

TITLE (PROVISIONAL)	Influences on university students' intention to receive recommended vaccines: a cross-sectional survey
AUTHORS	Landowska, Kate; Waller, Jo; Bedford, Helen; Rockliffe, Lauren; Forster, Alice

VERSION 1 - REVIEW

REVIEWER	Pauline Paterson London School of Hygiene & Tropical Medicine, UK
REVIEW RETURNED	30-Mar-2017

GENERAL COMMENTS	This article 'Influences on university students' intention to receive recommended vaccines' is the result of an interesting analysis of a large survey of students in London, UK, about which factors influence their intention to vaccinate with recommended vaccines. The findings from this study will be informative to the immunization community. I have a few minor suggestions which I detail below. The introduction would benefit by having a brief paragraph explaining the vaccines recommended to students and the importance of these vaccines. In the second paragraph of the methods, the authors state 'Items were developed from previous surveys'. Why were these surveys chosen? In the results, it would be useful to know the dates when students completed the survey (i.e. between when and when in 2015). It would also be interesting to know the breakdown of ethnicity and the breakdown of religion. The authors state 'approximately 50% had lived in the UK until at least the age of four, likely due to a high proportion of international students completing the survey'. Was this percentage similar to the proportion of international students at UCL? In the discussion, the authors highlight the Internet as the most common vaccine-related source for students. It may be worth highlighting past studies on uses of information source as well as trustworthiness of different information sources. For example, the paper by Boudier et al. (2015) 'Transparency in Europe: A Quantitative Study' in the journal Risk Analysis.
---

REVIEWER	Benjamin Kagina University of Cape Town South Africa
REVIEW RETURNED	01-May-2017

GENERAL COMMENTS	Elaboration of "No" answers in the Review Checklist: 3) The use of online questionnaires is a cost-effective method to collect survey data. However, piloting of questionnaires when using such an approach is very important to avoid incidences where there could be lack of clarity to some questions. Did the authors rule out the possibility of lack of clarity to some questions prior administering the questionnaires online? If not, can this be considered as another study limitation? 7) Lines 21-24 explains how a median split was used to create a binary outcome. How robust is the use of the median as reference value to generate two groups in this study? This can be investigated by authors using median absolute deviation (MAD) plus or minus the median and then assess if the univariate analysis is affected (i.e, sensitivity analysis of using the median as a split reference value). How accurate do the authors think the participants answered the question on previously having complete childhood immunisation? Related to this, the question on previous adverse reaction to vaccination may be inaccurately answered in circumstances where an adverse event happened in childhood and the students were not aware of as they were too young when this happened. Can these two issues be considered as additional limitations to the study?
--

VERSION 1 – AUTHOR RESPONSE

Reviewer: 1

Pauline Paterson

London School of Hygiene & Tropical Medicine, UK

Please state any competing interests or state 'None declared': None declared

Please leave your comments for the authors below This article 'Influences on university students' intention to receive recommended vaccines' is the result of an interesting analysis of a large survey of students in London, UK, about which factors influence their intention to vaccinate with recommended vaccines.

The findings from this study will be informative to the immunization community. I have a few minor suggestions which I detail below.

The introduction would benefit by having a brief paragraph explaining the vaccines recommended to students and the importance of these vaccines.

Thank you. The main vaccination recommended for UK university entrants is the Meningococcal ACWY vaccination. We have now provided further detail about this vaccination and its importance for students, and taken the opportunity to update the discussion in light of new data on uptake of the vaccination.

In the second paragraph of the methods, the authors state 'Items were developed from previous surveys'. Why were these surveys chosen?

These surveys were selected as they investigated predictors of vaccine acceptance (i.e. Theory of Planned Behaviour constructs) and positive behaviour change. We have now included this information in the manuscript.

In the results, it would be useful to know the dates when students completed the survey (i.e. between when and when in 2015). It would also be interesting to know the breakdown of ethnicity and the breakdown of religion.

Thank you. We have now included details about the period over which the study was conducted and provided a breakdown of participants' ethnicity and religion.

The authors state 'approximately 50% had lived in the UK until at least the age of four, likely due to a high proportion of international students completing the survey'. Was this percentage similar to the proportion of international students at UCL?

Thank you. The number of participants who lived outside the UK until the age of 4 was approximately 44%, slightly higher than the percentage of international students attending UCL, which is currently 35%. However, we have now discussed the over-representation of potentially non-UK born participants in the study as a limitation.

In the discussion, the authors highlight the Internet as the most common vaccine-related source for students. It may be worth highlighting past studies on uses of information source as well as trustworthiness of different information sources. For example, the paper by Boudier et al. (2015) 'Transparency in Europe: A Quantitative Study' in the journal Risk Analysis.

Thank you. We have now included additional references to studies looking at the use of different information sources and information trustworthiness.

Reviewer: 2

Benjamin Kagina

University of Cape Town, South Africa

Please state any competing interests or state 'None declared': None declared

Please leave your comments for the authors below Elaboration of "No" answers in the Review Checklist:

3) The use of online questionnaires is a cost-effective method to collect survey data. However, piloting of questionnaires when using such an approach is very important to avoid incidences where there could be lack of clarity to some questions. Did the authors rule out the possibility of lack of clarity to some questions prior administering the questionnaires online? If not, can this be considered as another study limitation?

Thank you for this comment. We agree that piloting the survey would have been a helpful way to establish the clarity of the questions. However, this study was conducted as part of an MSc project and due to time restrictions we were unable to do this. We have now discussed this as a limitation of the study.

7) Lines 21-24 explains how a median split was used to create a binary outcome. How robust is the use of the median as reference value to generate two groups in this study? This can be investigated by authors using median absolute deviation (MAD) plus or minus the median and then assess if the

univariate analysis is affected (i.e, sensitivity analysis of using the median as a split reference value).

Thank you for this suggestion. As the data were highly skewed we were unable to perform a sensitivity analysis using MAD plus the median as the splitting point, as we were already splitting at 2-9 v 10. We did however conduct an analysis using MAD minus the median (2-8 v 9-10), and it showed the same significant associations between the predictor and outcome variables as in the original analysis. We have now referred to this analysis in the manuscript and would be happy to provide the results as supplementary material, if requested by the editor.

How accurate do the authors think the participants answered the question on previously having complete childhood immunisation? Related to this, the question on previous adverse reaction to vaccination may be inaccurately answered in circumstances where an adverse event happened in childhood and the students were not aware of as they were too young when this happened. Can these two issues be considered as additional limitations to the study?

Thank you for these comments. We agree that there is the possibility that participants may have provided inaccurate responses to such questionnaire items. We have now discussed this as a limitation of the study. However, one could argue that childhood vaccination experience would only have an impact on vaccine attitudes and behaviour in young adults if they were able to remember it, so it is possible that their perceptions of childhood immunisation completion or adverse events may be more important than the actual events themselves.

VERSION 2 – REVIEW

REVIEWER	Pauline Paterson London School of Hygiene & Tropical Medicine, UK
REVIEW RETURNED	20-Jun-2017
GENERAL COMMENTS	Comments addressed - thank you.